# Knowledge, attitudes, and practices of cardiac rehabilitation and barriers to referral among cardiologists in Saudi Arabia: A cross-sectional survey

**Ahmed Mohammed Almoghairi**[1,2]*, **Jane O'Brien**[2], **Anna Doubrovsky**[2], **Jed Duff**[2,3]

**1** College of Nursing, Shaqra University, Saudi Arabia, **2** School of Nursing, Queensland University of Technology, Brisbane, Australia, **3** Royal Brisbane and Women's Hospital, Australia

* Ahmed@su.edu.sa

## Abstract

### Background

Cardiac rehabilitation (CR) is an effective secondary prevention intervention, yet it is globally underutilized. Physicians play a key role in CR uptake by eligible patients through encouragement and referral to the program. This study assessed the knowledge, attitudes, and practices concerning CR among cardiologists in the Kingdom of Saudi Arabia (KSA), identified barriers to patient referrals to CR programs, and proposed strategies to increase service adoption.

### Methods

We conducted an observational cross-sectional study in which an online questionnaire was distributed via email to cardiologists and cardiology fellows during the Saudi Heart Association's annual conference in October 2023 and through social media platforms. Participants were required to have at least six months of clinical practice in managing patients, including those with coronary heart disease (CHD) following percutaneous coronary intervention (PCI).

### Results

Of the 140 cardiologists surveyed, 106 completed more than 95% of the questionnaires. The cohort, which was primarily male (88.7%), included 67% consulting cardiologists, 15.1% fellows, and 17.9% specialists in areas such as general cardiology (29.2%), interventional cardiology (21.7%), and echocardiography (20.8%). Major barriers included a lack of local CR services (72.6%) and inadequate referral systems (41.5%). Despite the challenges and mixed views on the effectiveness of CR in KSA, attitudes toward CR were largely positive. The knowledge scores averaged 7.97, indicating a moderate to high understanding of CR services and benefits. Referral

**Data availability statement:** All relevant data are within the manuscript and its Supporting Information files.

**Funding:** The author(s) received no specific funding for this work.

**Competing interests:** The authors have declared that no competing interests exist.

practices vary widely and are influenced by demographic and workplace factors, mainly geographic location.

## Conclusions

While cardiologists in KSA generally have reasonable knowledge of CR and its benefits, substantial barriers hinder its broader implementation. There is enthusiasm for adopting diverse CR models; thus, further research is necessary to explore and evaluate alternative CR approaches, including home-based CR and telerehabilitation, to enhance patient care.

## 1. Introduction

Cardiovascular diseases (CVDs) represent a formidable global health challenge and are the primary cause of death and disability worldwide. In 2019, CVDs were responsible for an estimated 18.6 million deaths, representing 31% of global fatalities [1]. Over 80% of these deaths occur in developing countries, where the prevalence of CVDs has increased over the past three decades, from 4,624 to 7,769 cases per 100,000 individuals [2]. Coronary heart disease (CHD) remains the predominant cause of mortality among CVD patients, with a death rate of 112.37 per 100,000 reported in 2020 [3]. The Middle East, North Africa, Eastern European countries, and Central Asia are among the most impacted regions of the developing world [3]. In the Kingdom of Saudi Arabia (KSA), CVDs account for more than 45% of all fatalities, with CHD accounting for 23.1% of the total deaths nationwide [4,5].

The management of CVD places an extensive financial burden on healthcare systems [6]. Moreover, the higher rates of premature death associated with CVDs impose a significant economic impact at both the familial and national levels in developing nations [2]. Secondary prevention therapy, aimed at identifying and managing risk factors and controlling associated morbidities, can substantially mitigate this burden. Cardiac rehabilitation (CR) is a multifaceted process aimed at restoring individuals' optimal physical, mental, emotional, social, and economic well-being [7]. As a comprehensive, interdisciplinary secondary prevention strategy, CR has demonstrated efficacy in reducing mortality rates [8,9], enhancing health-related outcomes [8–12], and minimizing rehospitalization and overall treatment costs [8,13]. CR benefits have also been recognized in countries with low and middle incomes [8,14].

Comprehensive CR programs incorporate essential elements delivered in three phases of progression depending on the patient's diagnosis [15,16]. Phase I CR is administered during the inpatient period and involves patient education, early supervised physical exercise, and the management of risk factors [15,16]. Phase II, the early outpatient stage, commences upon the patient's discharge from the hospital and includes a supervised program of physical training, dietary management, risk factor control, and mental health support [15,16]. Phase III, the late stage of outpatient CR, focuses on maintaining the lifestyle modifications established in Phases I and II and encompasses regular reassessments of risk factors, medication adherence, and psychological well-being [15,16].

Enrollment in outpatient CR programs after a cardiac event or revascularization procedure is strongly recommended as a Class IA guideline by prominent international bodies such as the European Society of Cardiology, American Heart Association, and American College of Cardiology [7,9]. Unfortunately, CR programs are underused worldwide [8,17,18], and participation is less than 30% in high-income countries [17]. In low/middle-income countries, the availability of such programs is inadequate [19], and the utilization of these services, when available, is very poor [20,21]. Numerous studies, predominantly conducted in developed countries, have explored the factors contributing to this low uptake of CR programs and categorized them into three levels: patient, physician, and health system [22–24].

Despite the substantial burden imposed by the care of patients with CVDs in developing nations, there remains a notable dearth of data concerning the three-level factors leading to low participation rates [21], especially in KSA. Cardiologists are instrumental in encouraging patients to participate in CR, and their awareness and perspectives on these programs can substantially influence their referral practices, which are crucial for patients to initiate the program [25]. Therefore, this study aimed to assess the knowledge, attitudes, and practices of CR among cardiac specialists practicing in KSA. This study also aimed to identify the perceived barriers that could inhibit the referral of patients with CHD following percutaneous coronary intervention (PCI) to these programs. Strategies from international evidence to promote the uptake of CR services have also been proposed to solicit insights and recommendations for future program implementation within the country.

## 2. Materials and methods

### 2.1. Design and procedure

This study employed an observational cross-sectional survey design and used data from an online questionnaire designed via web-based survey software (Qualtrics, Qualtrics,2020, Provo, Utah, USA). The study was approved by the University Human Research Ethics Committee at the Queensland University of Technology (approval number: LR 2023-6924-14433). The initiation and completion of the electronic questionnaires were regarded as implied consent, thereby waiving the need for separate consent forms. The reporting of this study conformed to the Strengthening the Reporting of Observational Studies in Epidemiology (STROBE) guidelines for cross-sectional studies.

### 2.2. Participants

Cardiologists practicing in KSA who were involved in the care of patients with CHD after their PCI procedures constituted the target population for this study. Eligibility for inclusion was limited to cardiac-specialized physicians and registered cardiology fellows with a minimum of six months of experience in the field.

### 2.3. Recruitment

For participant recruitment, convenience sampling was used during the annual Saudi Heart Association Conference held in Riyadh from October 13th to 15th, 2023. This conference, the largest of its kind in the Gulf and Middle East, attracts hundreds of cardiologists each year. Participants were invited via email distributed by the event organizers, who detailed the study aims and objectives, the identities and affiliations of the researchers, the estimated time required to complete the survey, and assurances regarding the anonymity and security of the data. The email contained a link to the survey website where participants were required to enter their responses manually. Furthermore, over the subsequent two months, cardiologists were contacted through professional social media platforms, such as LinkedIn, and internal organizational communication channels, including WhatsApp and Telegram.

### 2.4. Variables and data collection

#### 2.4.1 Demographics and workplace characteristics.
This section of the survey collected data from cardiologists on their age, sex, nationality, specialty, degree of specialty, country of graduation, and years of clinical experience, as well as their workplace type, size, location, and approximate number of visits by patients treated with PCI.

**2.4.2 Knowledge, attitudes, practices and barriers to cardiac rehabilitation.** A structured self-administered knowledge, attitude, and practice (KAP) questionnaire was used to collect information on the current practices of secondary CHD prevention in KSA. This KAP instrument was developed, utilized, and validated by researchers and field experts from Lebanon, who based their items on a comprehensive literature review [26]. The questionnaire, consisting of 25 items, was derived from the three primary elements of the KAP framework as outlined by the WHO [26], with an additional component addressing the barriers physicians face when referring patients.

The survey was divided into five distinct parts, each focusing on a specific subject area. The first section evaluated participants' self-assessed knowledge of CR and its key components, asking cardiologists to categorize their level of understanding (excellent, good, medium, poor, and very poor). The second part explored cardiologists' perceptions of the value and effectiveness of CR in the secondary prevention of CHD post-PCI, with participants rating their attitudes on a scale from 1 (strongly agree) to 5 (strongly disagree). The third section highlights the current practices of physicians in the secondary prevention of CHD. The fourth section probed the barriers that might hinder physicians' willingness or ability to refer patients to CR programs. Finally, the questionnaire was completed by soliciting cardiologists' opinions on the feasibility and acceptability of the various CR models in KSA.

## 2.5. Bias

Cross-sectional studies are vulnerable to selection bias, primarily due to potential discrepancies between responders and non-responders, which is known as nonresponse bias. Additionally, sampling bias may have occurred, as it is anticipated that volunteer responders differ systematically from the population in terms of their sociodemographic characteristics and attitudes. The reliance on self-reported data in this survey may also introduce recall bias, affecting the accuracy of the responses regarding referral practices.

## 2.6. Data analysis

Participant survey data were entered into R (version 4.3.1) [27] and RStudio (Version 2023.09.1) [28] for analysis. Descriptive statistics, including participant demographics, workplace characteristics, cardiology practices, and responses to questions regarding barriers to and facilitators of patient referral to cardiac rehabilitation following PCI, were used to characterize the survey sample. Attitudes were assessed via five items on a Likert scale ranging from "strongly disagree" to "strongly agree." Knowledge was evaluated via two questions on a scale ranging from "very poor" to "excellent." Cardiac rehabilitation referral practices were assessed via a single question on the proportion of patients referred for cardiac rehabilitation after PCI. Given that more than one-third of cardiologists reported an inability to refer patients to cardiac rehabilitation, this question was analyzed as a binary variable (yes/no), indicating whether any patient was referred.

Univariate linear regression was employed to examine the relationships between cardiologists' total knowledge (summed scores from the two questions) and descriptive variables (demographic, workplace, and other variables). A multivariate model was subsequently constructed from the univariate results, including variables with a significance level of $P < 0.05$, and a backward step procedure was used to build the model. Barrier and facilitator variables were not included to avoid overfitting. A complete case analysis was conducted owing to the rarity of the missing data.

Logistic regression was used to explore the relationships between cardiologists' referral practices for patients following PCI and cardiac rehabilitation (yes/no) and descriptive variables. A multivariate model was developed from the univariate results, incorporating variables with a significance level of $P < 0.1$ and using a backward-step procedure to build the model. Barrier and facilitator variables were not included to avoid overfitting. A complete case analysis was performed, excluding seven participants because of missing data. The model assumptions for both regression analyses were evaluated and found not to be violated. Statistical significance was set at the 5% level ($P < 0.05$).

## 3.  Results

### 2.5  Participants

According to the most recent Statistical Yearbook published by the Ministry of Health in KSA, the total number of cardiologists practicing across all health sectors by 2022 was 2,125 specialists. A total of 140 cardiologists participated, with 106 completing at least 95% of the questionnaires (88.7% male). Notably, 34 participants (24.3%) initiated the survey but did not proceed beyond the second page. Physicians who probably face significant time pressures and are not compensated for their participation in surveys represent the entirety of the sample.

### 2.6  Participant demographics

Table 1 summarizes the demographics of the participants. Most survey respondents (73.6%) were aged 35--55 years, with a predominantly male cohort ($n = 94$; 88.7%). Almost 60% ($n = 63$) of the participants were Saudi Arabian citizens, 67.0% ($n = 71$) were consulting cardiologists, 15.1% were cardiology fellows, and 17.9% were specialist cardiologists. The participants specialized in general cardiology (29.2%), interventional cardiology (21.7%), and echocardiography (20.8%), with few in critical care or cardiac rehabilitation (<5%). One-third of the surveyed cardiologists had received education in KSA (32.1%). Furthermore, 18.9% had received their education in Canada, and 12.3% had received their education in other countries within the Middle East. Fewer than 10% had received their education in Europe, the United States of America, or South Asia. More than 14% did not report where they had received education. Half of the participants had 5–15 years of medical practice, and almost a quarter had more than 20 years of medical practice experience.

### 2.7  Participant workplace characteristics

Most participating cardiologists worked in the public sector (78.3%), as shown in Table 2. Nearly 10.4% worked within the private sector, and 11.3% worked in both sectors. Workplace sizes varied from 0 beds to more than 1000 beds, but most physicians worked in health facilities with fewer than 300 beds (54.7%). Most respondents (90.6%) worked with CHD-diagnosed patients who underwent PCI. The number of PCI-treated patients varied from one-third of cardiologists who only saw 0–10 patients per week to 10% who saw more than 40 patients per week. Most participants practiced in the central region of KSA, which includes the capital Riyadh (54.7%). Another quarter was found in the western (17.9%) and eastern (10.4%) regions. Few cardiac-specialized doctors were located in the northern (1.9%) or southern region (8.5%). Seven cardiologists did not specify the city or region where they worked.

### 2.8  Barriers and facilitators to the referral of patients following percutaneous coronary intervention to cardiac rehabilitation

Participants were surveyed on the barriers they saw to refer patients with CHD to CR following PCI and who should take the initiative to implement this service. For these questions, the participants could choose more than one option. The lack of access to specialist CR services in their area was most significant, with 72.6% ($n = 77$) of participants selecting this barrier. A lack of local systems to support CR referrals was cited by 41.5% ($n = 44$) of the respondents. Approximately one-third of the cardiologists reported a lack of knowledge about CR services in my area (28.3%, $n = 30$) and patient factors, such as the cost of care and distance (32.1%, $n = 34$). Almost 10% (9.4%, $n = 10$) had no barriers in referring patients to CR.

Cardiologists were asked "who do they think should take the initiative to implement outpatient CR programs in Saudia Arabia?". Most (59.4%, $n = 63$) reported that all professional care providers should, followed by the Ministry of Health (52.8%, $n = 56$) and physicians (52.8%, $n = 56$). Fewer cardiologists were recorded by insurance companies (44.3%, $n = 47$) or policymakers (39.6%, $n = 42$). In a free-text response, participants were also asked what they considered a priority for the implementation of a future CR program in Saudi Arabia. A schematic representation of the variety of responses

**Table 1. Demographics.**

| | Overall (N = 106) |
|---|---|
| **Sex** | |
| Male | 94 (88.7%) |
| Female | 12 (11.3%) |
| | |
| **Age** | |
| < 35 | 15 (14.2%) |
| 35-40 | 34 (32.1%) |
| 41-55 | 44 (41.5%) |
| 56-65 | 12 (11.3%) |
| > 65 | 1 (0.9%) |
| | |
| **Nationality** | |
| Saudi | 63 (59.4%) |
| Non-Saudi | 43 (40.6%) |
| | |
| **Specialty Level** | |
| Cardiology Fellow | 16 (15.1%) |
| Consultant Cardiologist | 71 (67.0%) |
| Specialist Cardiologist | 19 (17.9%) |
| | |
| **Specialty Area:** | |
| General Cardiology | 31 (29.2%) |
| Interventional Cardiology | 23 (21.7%) |
| Echocardiography | 22 (20.8%) |
| Electrophysiology | 9 (8.5%) |
| Cardiac Surgery | 5 (4.7%) |
| Critical Care Cardiology | 4 (3.8%) |
| Cardiac Rehabilitation Specialty | 3 (2.8%) |
| Other/Mixed | 9 (8.5%) |
| | |
| **Education Location** | |
| Saudi Arabia | 34 (32.1%) |
| Canada | 20 (18.9%) |
| Middle East/Africa, not Saudia Arabia | 13 (12.3%) |
| Europe (UK, France, Germany, Poland) | 10 (9.4%) |
| USA | 8 (7.5%) |
| South Asia | 3 (2.8%) |
| Multiple | 3 (2.8%) |
| Missing | 15 (14.2%) |
| | |
| **Years in medical practice** | |
| < 5 years | 17 (16.0%) |
| 5-10 years | 23 (21.7%) |
| 10-15 years | 30 (28.3%) |
| 15-20 years | 12 (11.3%) |
| >20 years | 24 (22.6%) |

is presented in Fig 1. The participants recommended staff education and patient coaching. They suggested that rehabilitation should be timed early before hospital discharge and that it should be resourced through facilities with specific programs that include policies, protocols, and guidelines. The lack of availability of CR services and proper financing are highlighted. Finally, participants recognized that CR was required for better patient outcomes.

## 2.9. Cardiologist attitudes

The participants were asked five questions related to their attitudes toward CR on a five-point Likert scale ranging from strongly disagree (score = 1) to strongly agree (score = 5). The questions and responses are shown in Fig 2a. The participants were positive and agreed to more than 90% of the four attitudes; however, they did not agree when they were asked if CR within KSA was effective. The Cronbach's alpha for these five questions was poor at 0.45; thus, the items were treated separately through further analyses.

## 2.10. Cardiologist knowledge

Cardiologists were asked two questions regarding their knowledge of CR in improving patient outcomes and core services, as shown in Fig 2b. Both questions were ranked from very poor (score = 1) to excellent (score = 5). Together, they had a Cronbach's alpha of 0.753 and were thus summed to produce a total knowledge score with a possible range of 2--10, with a mean of 7.97 (SD 1.437).

Univariate linear regression was used to determine which demographic and workplace characteristics plus barrier and facilitator responses influenced the total knowledge score (see S1 Table. Variables with a significance of less than $p < 0.05$ in the univariate models were added to a multivariable linear regression model. These were sex (male, female), attitudes (Likert scores of 1--5), workplace type (public, private, or both), management of patients following PCI (yes, no), and referral of patients to CR (yes/no). A backward methodology was used to select the final variables in the linear regression model (Table 3, Model 1). The final model had four predictors that explained 19% of the variance (adjusted $R^2 = 0.197$, $F = 7.015$, $P < 0.001$). The results revealed that females had almost one-unit higher total knowledge score than males did ($\beta = 0.896$, $P = 0.027$), with higher agreement with one attitude question (Do you think that your patients' outcomes improved when they were enrolled in CR?) The results revealed an increase of 0.68 in the total knowledge score per step of the attitude Likert scale ($\beta = 0.681$, $P < 0.001$); those participants who managed patients after PCI were predicted to have higher knowledge scores of more than one unit than those who did not ($\beta = 1.309$, $P = 0.012$), and cardiologists who referred any patient to CR had a greater knowledge score of 0.6 units higher ($\beta = 0.602$, $P = 0.033$).

## 2.11. Cardiologist practice

Cardiologists were surveyed on their current practice with patients who underwent PCI procedures, and the results are shown in Table 4. More than half of the participants (52.8%) chose to quit smoking, followed by more than a quarter who recommended a CR program (28.3%). Thirty percent of the cardiologist participants reported that CR was not an option available to their patients, either they did not refer them or they could not refer them. However, 37.8% referred 50% or fewer of their PCI-treated patients to CR, and 25.5% sent more than 50% of their patients to the program. Half of the cardiologists wanted their patients to start CR in hospital settings before discharge, with another 35.6% starting shortly after discharge. Fifteen percent indicated that they would not prescribe CR to their patients who underwent PCI. Most of the participating cardiologists reported that it was difficult to refer patients to the CR in KSA (74.5%), with only 10.4% reporting that it was easy.

## 2.12. Practice: Referral of patients to cardiac rehabilitation (n = 99)

We analyzed whether any demographic or workplace characteristics or barrier and facilitator responses influenced whether the participants (*n* = 99) referred patients to cardiac rehabilitation (yes/no) through univariate logistic regression

**Table 2. Workplace characteristics.**

| | Overall (N = 106) |
|---|---|
| **Workplace type** | |
| Government sector | 83 (78.3%) |
| Private Sector | 11 (10.4%) |
| Both | 12 (11.3%) |
| | |
| **Workplace Size** | |
| | |
| 0-99 beds | 28 (26.4%) |
| 100-299 beds | 30 (28.3%) |
| 300-499 beds | 26 (24.5%) |
| 500-999 beds | 14 (13.2%) |
| ≥ 1000 beds | 8 (7.5%) |
| | |
| **Manage patients post-PCI** (Yes) | 96 (90.6%) |
| | |
| **Number of PCI treated patients seen per week** | |
| 0-10 patients | 40 (37.8%) |
| 11-20 patients | 29 (27.4%) |
| 21-30 patients | 17 (16.0%) |
| 31-40 patients | 7 (6.6%) |
| > 40 patients | 13 (12.3%) |
| | |
| **Workplace Region** | |
| Central region (including Riyadh $n = 51$) | 58 (54.7%) |
| Eastern region | 11 (10.4%) |
| Northern region | 2 (1.9%) |
| Southern region | 9 (8.5%) |
| Western region (including Jeddah $n = 13$) | 19 (17.9%) |
| Other, Unspecified | 7 (6.6%) |

(see S2 Table). Seven participants did not fill out the practice question and were thus excluded. Variables with a significance of less than $P = 0.1$ in the univariate models were added to a multivariable logistic regression model. These included Saudi nationality (yes, no), educational location, total knowledge score, and attitude score (Do you think that cardiac rehabilitation in Saudi Arabia is effective? Likert scale from 1--5) and workplace region (recategorized into central versus southern, western, eastern, and northern regions).

A backward methodology was used to select the final variables in the model (**Table 3**, Model 2). The logistic regression model was statistically significant ($\chi 2 = 19.407$, $P < 0.001$) and explained 24.9% (Nagelkerke's R2) of the variance in the model. Only four variables remained statistically significant in the final model. The workplace region, where the central region was 3.3 times more likely to refer patients to the CR, than the southern, western, eastern, and northern regions were (OR 3.27, 95% CI 1.27--9.03). Saudi nationals were 2.7 times more likely to refer patients for CR than non-Saudi cardiologists were (OR 2.74 95% CI 1.05--7.58); the total knowledge score, where participants were 45% more likely to

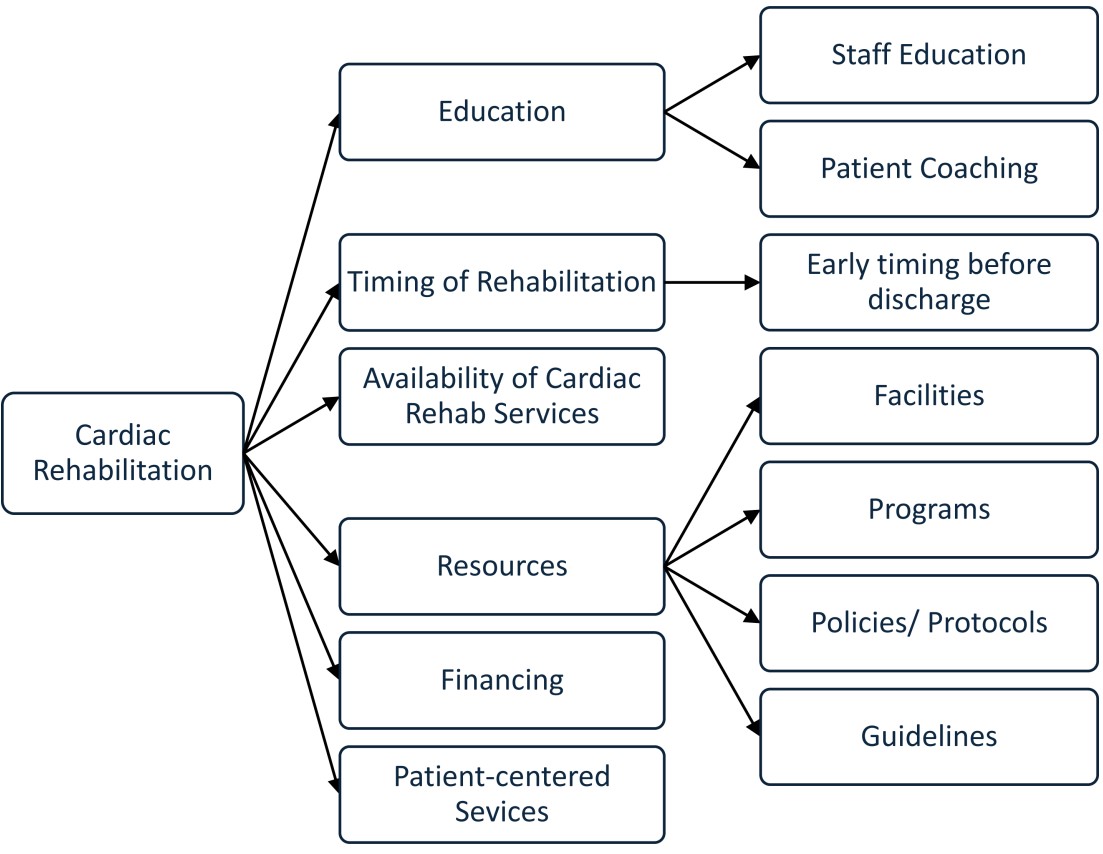

**Fig 1. Concept map.**

refer patients for CR per score unit (OR 1.45 95% CI 1.04--2.06 per unit of total knowledge score), and a positive attitude toward "Cardiac rehabilitation in Saudi Arabia is effective" resulted in higher referral (94% more per Likert step) of patients to CR (OR 1.94 95% CI 1.20--3.32 per Likert step from 1--5).

## 4. Discussion

This study revealed low referral rates for CR among cardiologists in KSA, primarily due to the limited availability of CR programs and systemic barriers, such as inadequate referral systems. However, positive attitudes toward CR suggest opportunities for improvement through targeted interventions such as home-based CR (HBCR). This study is the first national survey in KSA to assess the awareness, perceptions, and practices of cardiology specialists regarding CR for patients with CHD post-PCI and identify factors influencing their referral decisions. Nevertheless, a similar cross-sectional study conducted in KSA recently investigated the attitudes of physicians, including internal medicine specialists, general physicians, and pulmonologists, toward prescribing pulmonary rehabilitation programs for patients with chronic obstructive pulmonary disease, identifying factors influencing their referral practices [30]. Additionally, a simultaneous survey in the same year evaluated the perspectives of physiotherapists in providing CR to patients with heart failure (HF), uncovering barriers that affect their referrals [31].

Our results reveal a significant gap in referral practices, with 63.3% of cardiologists reporting that they refer eligible patients with CHD to CR programs at varying rates: 50% or less by 37.8% and 50% or more by 25.4%. Conversely, one-third of the participating cardiologists were reluctant to refer patients at all. This low referral rate is corroborated by

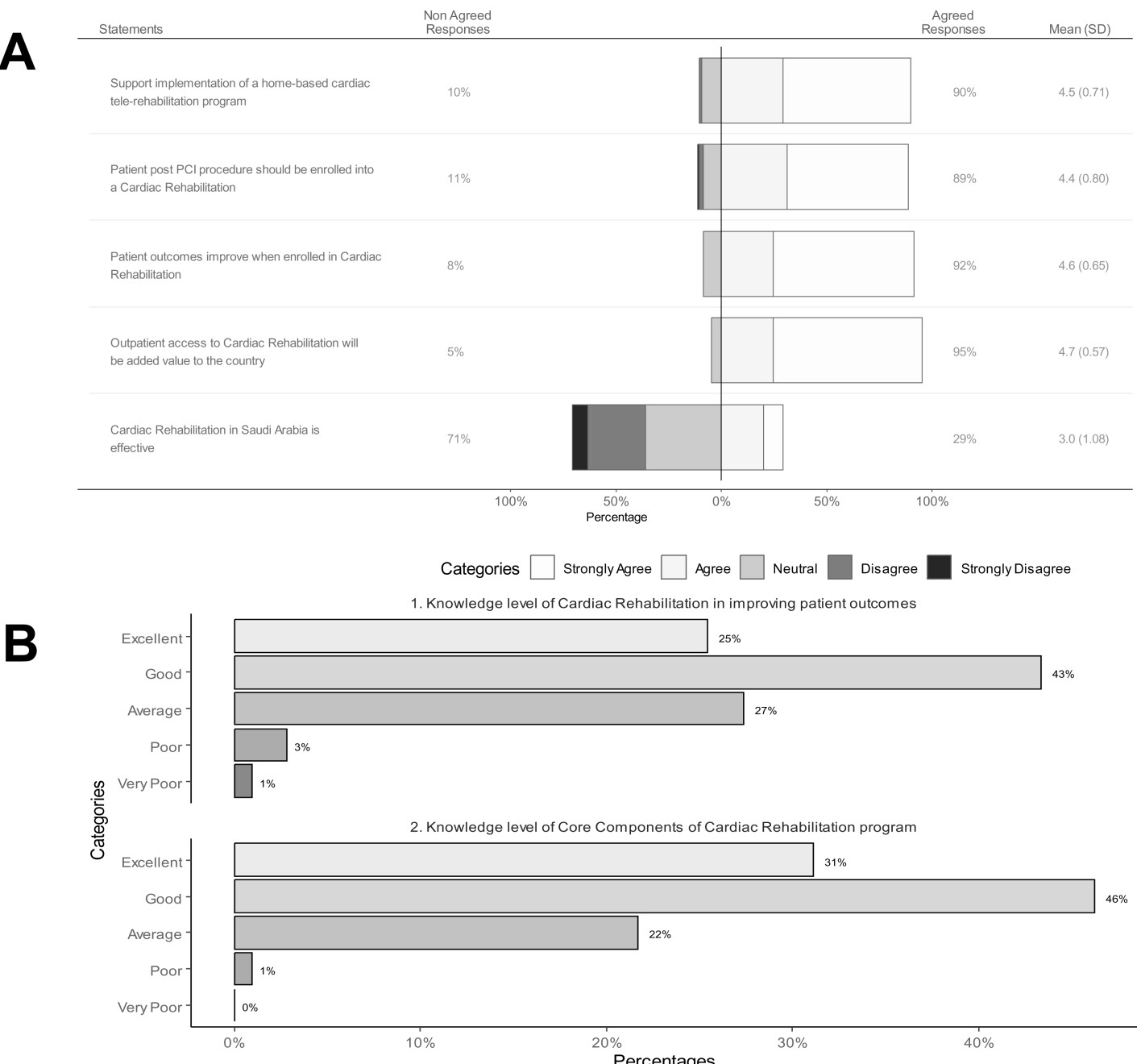

**Fig 2. Knowledge and attitude responses.**

**Table 3. Multivariable models.**

| Linear Regression Model (Model 1) Total Knowledge score dependent variable *n*=99 | | | | | | | |
| --- | --- | --- | --- | --- | --- | --- | --- |
| Independent variables | Estimate | Estimate 2.5% limit | Estimate 97.5% limit | Std. Error | t value | *P* | Regression Summary |
| (Intercept) | 3.134 | 0.911 | 5.357 | 1.120 | 2.799 | 0.006 | *Adjusted R²=0.197 F=7.015 P<0.001* |
| Sex (Female) | 0.896 | 0.103 | 1.689 | 0.399 | 2.243 | 0.027 | |
| Attitude (Scale 1–5) *Do you think that your patients' outcomes improved when they are enrolled in cardiac rehabilitation* | 0.681 | 0.271 | 1.091 | 0.207 | 3.298 | 0.001 | |
| Manage Patients Following PCI (Yes) | 1.309 | 0.298 | 2.320 | 0.509 | 2.571 | 0.012 | |
| Referred any patients to Cardiac Rehabilitation (Yes) | 0.602 | 0.050 | 1.154 | 0.278 | 2.166 | 0.033 | |
| Logistic Regression Model (Model 2) Refer post-PCI patients to cardiac rehabilitation (Yes/No) dependent variable *n*=99 | | | | | | | |
| Independent variables | Estimate | Std. Error | t value | *P* | OR | 95%CI OR | Regression Summary |
| (Intercept) | -5.242 | 1.754 | -2.988 | 0.003 | 0.005 | 0.001-0.135 | *Nagelkerke R²=0.249 χ²=19.407 P<0.001* |
| Total Knowledge Score | 0.369 | 0.172 | 2.145 | 0.032 | 1.446 | 1.040-2.056 | |
| Attitude (Scale 1–5) *Do you think that cardiac rehabilitation in Saudi Arabia is effective?* | 0.662 | 0.257 | 2.580 | 0.010 | 1.938 | 1.202-3.317 | |
| Saudi Nationality (Yes) | 1.007 | 0.501 | 2.012 | 0.044 | 2.738 | 1.047-7.583 | |
| Region *Central region including Riyadh compared to other regions (Southern, Western, Eastern and Northern)* | 1.184 | 0.497 | 2.384 | 0.017 | 3.267 | 1.268-9.026 | |

a recent study that examined 104 Saudi patients with CHD who underwent PCI at a prominent cardiac center in Riyadh. This study aimed to assess adherence to secondary prevention measures and identify barriers to CR program enrollment. Only 11 patients (10.6%) had been referred to a CR program, with 88% citing the absence of physician support as a major obstacle to accessing CR services [32].

Most of the specialized physicians who participated in our survey reported difficulties in the referral process, primarily because of limited access to specialist CR services in their regions. This challenge is mirrored in the findings of Al dhaihir et al. (2022) [31], who surveyed 553 physiotherapists across KSA and reported that 59.9% perceived the scarcity of available programs as the principal barrier affecting referrals to CR for patients with HF. This issue of low program availability is a global concern, with fewer than 25% of developing countries providing CR services to their populations [21]. In KSA, the situation is particularly critical, with only five active governmental CR centers, three of which are located in Riyadh and two in the country's eastern region [33]. This distribution contributes to the observed regional disparities in referral practices in our study, with physicians located in the western or southern provinces being less likely to refer patients than their counterparts in the central or eastern parts of the country were. Research has demonstrated that a wider geographic distribution of CR services is correlated with higher referral rates [34]. The maldistribution of healthcare facilities mainly affects patients in remote areas, who are less likely to enroll in CR programs owing to geographic barriers [34].

The delivery of CR in KSA is significantly hindered by a lack of integration across various levels of cardiac care, stemming from the nation's fragmented healthcare system. The healthcare system in KSA is divided into two main sectors: the governmental sector, which provides free services to citizens, and the private sector, which operates on a fee-for-service basis [35]. The Ministry of Health directly manages and funds 60% of governmental healthcare institutions, while the remaining 40% fall under the jurisdiction of various other agencies, including military and educational institutions [35]. This

**Table 4. Practice for post-PCI patients.**

| | Overall (N = 106) |
|---|---|
| **Post-procedure recommendations** | |
| Do nothing, to be at rest | 3 (2.8%) |
| Exercise a bit | 16 (15.1%) |
| Quit smoking if they were smokers | 56 (52.8%) |
| See a therapist if they need mental health support | 1 (0.9%) |
| Start to attend a cardiac rehabilitation program | 30 (28.3%) |
| | |
| **Referrals to cardiac rehabilitation** | |
| Nil (0%) | 21 (19.8%) |
| Service not available | 11 (10.4%) |
| 1-25% | 22 (20.8%) |
| 25-50% | 18 (17.0%) |
| 50-75% | 9 (8.5%) |
| 75-99% | 6 (5.7%) |
| 100% | 12 (11.3%) |
| Unknown/Missing | 7 (6.6%) |
| | |
| **Preferred time to start cardiac rehabilitation** | |
| Starting in the hospital settings | 52 (49.1%) |
| Directly after discharge in their first visit | 26 (24.5%) |
| 4 weeks or more after their discharge | 12 (11.3%) |
| Would not prescribe it | 16 (15.1%) |
| | |
| **Difficulties of referral to cardiac rehabilitation (missing = 1)** | |
| Extremely difficult | 40 (37.7%) |
| Somewhat difficult | 39 (36.8%) |
| Neither easy nor difficult | 15 (14.2%) |
| Somewhat easy | 7 (6.6%) |
| Extremely easy | 4 (3.8%) |

fragmentation is evident in our findings, where 74.5% of physicians reported difficulties in issuing CR referrals, along with significant delays in referrals reaching eligible patients, who struggle to navigate the healthcare system [32].

The situation is further complicated by poor coordination and communication between cardiologists in provinces with limited cardiology care and CR providers, who are primarily concentrated in major cardiac centers within only two regions (Central and Eastern). Interviews with directors of cardiac centers and CR coordinators in KSA provided valuable insights into the factors contributing to the delayed expansion and uneven distribution of CR services across the country [36]. The qualitative analysis revealed widespread recognition of CR and its benefits among healthcare leaders, accompanied by significant concerns regarding the inadequacy of the current infrastructure to meet national demand [36]. According to policymakers, this challenge is primarily attributed to limited resources, particularly insufficient funding and a shortage of CR-trained personnel [36].

Our results further indicate that cardiologists might preferentially refer patients who they perceive as more likely to engage in treatment, with a third acknowledging that patient-related factors such as financial costs and long commuting distances influence their referral decisions. For example, a cohort study conducted in Sweden involving 31,297 patients with CHD referred to CR revealed that a commuting distance greater than 16 km significantly increased the likelihood of

nonattendance [37]. Similarly, in Iran, patients who were required to travel more than 30 minutes to a CR service site had lower participation rates [38]. Patients with CHD in KSA also cited distance to CR sites as a major personal barrier, particularly for 69% of those living in rural areas [32].

The successful implementation of alternative models, such as HBCR, has the potential to significantly extend service reach, particularly for patients who are less likely to participate in hospital-based programs [38]. The cardiologists who participated in our survey strongly endorsed the implementation of HBCR models within the country. This aligns with the findings of the survey conducted by Aldhahir et al. (2022), which reported that 74% of cardiac physicians across KSA preferred delivering the HBCR model to patients with HF [29]. Additionally, Saudi patients with CHD who had undergone PCI demonstrated a marked preference for HBCR, with 58.7% favoring it over the hospital-based model (17.3%) or hybrid model (11.5%) [32]. This preference was even more pronounced among individuals residing in rural areas (78.4%) than among those in urban settings (59.3%) [32]. The strong endorsement of the HBCR highlights its potential to overcome geographic and systemic barriers, particularly in rural regions, where traditional CR facilities are inaccessible.

The home-based model has proven effective in overcoming several barriers to CR accessibility, addressing challenges such as limited facilities, geographic constraints, transportation difficulties, financial burdens, and scheduling conflicts [39]. Globally, HBCR has been shown to be as safe and effective as traditional hospital-based CR for patients at low to moderate risk following revascularization, offering comparable costs while enhancing participation and adherence rates [40,41]. In KSA, a recent three-arm RCT—the first conducted in the country and the Gulf region—demonstrated the feasibility and suitability of HBCR in the local context, showing it to be as effective as outpatient CR and superior in maintaining post-intervention improvements in patients with CHD following coronary artery bypass grafting [42]. Additionally, patient preferences and acceptance must be considered, particularly in terms of digital literacy, cultural factors, and willingness to engage in remote rehabilitation. The successful and sustainable expansion of HBCR programs in KSA further depends on strong policy support, the development of viable financial models, and ongoing professional training and education [36]. Despite its potential, successful implementation of HBCR in KSA also requires addressing infrastructure challenges, such as the availability of telemonitoring technologies, trained personnel, and home-based medical equipment.

Administrative challenges, such as the absence of standardized criteria for patient eligibility and clear pathways for accessing CR programs, significantly hinder patient engagement [25]. Physicians identified the lack of local systems to support referrals as a substantial barrier affecting their referral practices, with almost half of them noting this as an issue. The integration of effective referral systems, such as the automation of referrals from discharge order sets or electronic records, is suggested to increase both referral rates and enrollment in CR programs [39]. This claim is supported by a retrospective cohort study in Canada that analyzed the registry and CR databases from 1996--2016. Employing interrupted time series analysis, the study evaluated the impact of automated referrals implemented in 2007 on the referral rates of eligible patients with CHD. After implementation, there was a notable increase in referral rates from 39.5% to 75.0% ($P<0.001$), underscoring the effectiveness of automation in improving CR referral processes [41].

Most participants expressed moderate-to-good knowledge about CR attributes and their benefits in patients treated with PCI. Our findings revealed greater CR referral practices among cardiologists with higher knowledge levels. Additionally, one-third of the respondent cardiac physicians reported that inadequacy of awareness about the programs led to their insufficient CR endorsement. Similarly, in Lebanon, one-third of cardiologists demonstrated a moderate level of knowledge about CR content (31.3%), while an equal proportion presented good to excellent knowledge (31.3%) and 39.9% to 43.3% regarding program benefits [32]. Despite generally positive attitudes toward CR among our study's respondents, two-thirds of the participants doubted the efficacy of local programs, which correlated with their reduced likelihood of referring patients. This contrasted with the remaining third, who believed in their effectiveness and showed a significantly greater inclination to refer. A similar trend was observed in Iran, where a qualitative study employing focus group discussions and conventional content analysis revealed that cardiologists' hesitancy to refer to CR was often due to insufficient knowledge of the program and uncertainties regarding its effectiveness in treatment [43].

### 4.1. Limitations

Our study had several limitations that merit careful consideration. The cross-sectional design, which relied on a convenience sampling strategy, limited the recruitment base and restricted data collection to a short period, potentially introducing selection bias. The relatively small and predominantly male composition of our sample size further complicates the extrapolation of our results to a broader population of cardiologists in KSA. This sample limitation also hindered our ability to apply statistical techniques that could adjust for confounding demographic and workplace characteristics. Additionally, 34 participants failed to complete the survey and were excluded, which could have introduced bias in the results. The incomplete responses and overall low response rate are likely attributable to the online administration of the survey, exacerbated by participants' professional constraints. Nonetheless, the use of an electronic questionnaire enhanced participant recruitment, allowing a broader survey.

## 5. Conclusion

This study assessed cardiologists' knowledge, attitudes, and practices regarding CR in KSA, identifying key barriers such as limited program availability and inadequate referral systems. While specialized physicians generally reported moderate to good knowledge of CR and its benefits for patient health, nearly one-third cited a lack of comprehensive understanding as detrimental to their treatment planning. Other impediments included the limited availability of CR programs, the absence of standardized referral systems, and patient personal reasons. Despite these challenges, participants displayed a positive attitude toward comprehensive secondary prevention programs and the adoption of diverse models within local contexts. However, two-thirds of the respondents expressed concerns about the quality of CR service delivery in the country. Referral behaviors varied, with those with higher knowledge levels and confidence in existing CR services and those working in the Central and Eastern provinces being more likely to recommend CR to their patients. Future research should include comprehensive randomized trials involving all levels of cardiology specialists in KSA, including graduate medical students enrolled in cardiology fellowships, to evaluate their KAP regarding CR. Additionally, research should prioritize the development and implementation of HBCR models and automated referral systems to enhance CR accessibility, particularly in underserved regions.

## Supporting information

**S1 Table. Knowledge Univariate Linear Regression n = 106.**
(DOCX)

**S2 Table. Practice Univariate Logistic Regression n = 99 (missing = 7).**
(DOCX)

**S3. STROBE Reporting Guidelines**
(DOCX)ACKNOWLEDGMENTS

The authors would like to thank the Deanship of Scientific Research at Shaqra University for supporting this work.

## Author contributions

**Conceptualization:** Ahmed Mohammed Almoghairi, Jane O'Brien, Anna Doubrovsky, Jed Duff.

**Data curation:** Ahmed Mohammed Almoghairi, Anna Doubrovsky, Jed Duff.

**Formal analysis:** Ahmed Mohammed Almoghairi, Jane O'Brien, Anna Doubrovsky, Jed Duff.

**Investigation:** Ahmed Mohammed Almoghairi.

**Methodology:** Ahmed Mohammed Almoghairi, Jane O'Brien, Anna Doubrovsky, Jed Duff.

**Project administration:** Ahmed Mohammed Almoghairi, Jane O'Brien, Jed Duff.

**Resources:** Jed Duff.

**Software:** Anna Doubrovsky.

**Supervision:** Jane O'Brien, Jed Duff.

**Validation:** Ahmed Mohammed Almoghairi, Jane O'Brien.

**Visualization:** Ahmed Mohammed Almoghairi, Jed Duff.

**Writing – original draft:** Ahmed Mohammed Almoghairi.

**Writing – review & editing:** Ahmed Mohammed Almoghairi, Jane O'Brien, Anna Doubrovsky, Jed Duff.

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
