## [Decision Letter · Decision Letter 0]

19 Feb 2025

PONE-D-25-01650Knowledge, Attitudes, and Practices of Cardiac Rehabilitation and Barriers to Referral among Cardiologists in Saudi Arabia: A Cross-Sectional SurveyPLOS ONE

Dear Dr. Almoghairi,

Thank you for submitting your manuscript to PLOS ONE. After careful consideration, we feel that it has merit but does not fully meet PLOS ONE’s publication criteria as it currently stands. Therefore, we invite you to submit a revised version of the manuscript that addresses the points raised during the review process.

We look forward to receiving your revised manuscript.

Kind regards,

Shukri AlSaif

Academic Editor

PLOS ONE

Reviewers' comments:

Reviewer's Responses to Questions

**Comments to the Author**

1. Is the manuscript technically sound, and do the data support the conclusions?

Reviewer #1: Yes

2. Has the statistical analysis been performed appropriately and rigorously? 

Reviewer #1: Yes

3. Have the authors made all data underlying the findings in their manuscript fully available?

Reviewer #1: Yes

4. Is the manuscript presented in an intelligible fashion and written in standard English?

Reviewer #1: Yes

5. Review Comments to the Author

Reviewer #1: The discussion should further explore the reasons behind regional disparities in referral rates, considering infrastructure, policy differences, and healthcare accessibility.

The study's reliance on convenience sampling introduces potential selection bias, which should be discussed in more detail along with recommendations for future randomized studies.

The section on home-based cardiac rehabilitation (HBCR) should provide more details on practical implementation in Saudi Arabia, considering infrastructure, patient preferences, and feasibility.

6. PLOS authors have the option to publish the peer review history of their article (what does this mean? ). If published, this will include your full peer review and any attached files.

**Do you want your identity to be public for this peer review?** For information about this choice, including consent withdrawal, please see our Privacy Policy .

Reviewer #1: **Yes: ** Nasser M Alorfi

---

## [Author Response · Author response to Decision Letter 0]

24 Mar 2025

We have uploaded a 'Response to Reviewers' file which include detailed answers to address each reviewer's comment.

Thank you

---

## [Decision Letter · Decision Letter 1]

13 Apr 2025

Knowledge, Attitudes, and Practices of Cardiac Rehabilitation and Barriers to Referral among Cardiologists in Saudi Arabia: A Cross-Sectional Survey

PONE-D-25-01650R1

Dear Dr. Almoghairi,

We’re pleased to inform you that your manuscript has been judged scientifically suitable for publication and will be formally accepted for publication once it meets all outstanding technical requirements.

Kind regards,

Shukri AlSaif

Academic Editor

PLOS ONE

Additional Editor Comments (optional):

Reviewers' comments:

Reviewer's Responses to Questions

**Comments to the Author**

1. If the authors have adequately addressed your comments raised in a previous round of review and you feel that this manuscript is now acceptable for publication, you may indicate that here to bypass the “Comments to the Author” section, enter your conflict of interest statement in the “Confidential to Editor” section, and submit your "Accept" recommendation.

Reviewer #1: All comments have been addressed

2. Is the manuscript technically sound, and do the data support the conclusions?

Reviewer #1: Yes

3. Has the statistical analysis been performed appropriately and rigorously? 

Reviewer #1: Yes

4. Have the authors made all data underlying the findings in their manuscript fully available?

Reviewer #1: Yes

5. Is the manuscript presented in an intelligible fashion and written in standard English?

Reviewer #1: Yes

6. Review Comments to the Author

Reviewer #1: All comments were addressed, paper is ready to be published in PLOSONE. The authors successfully addressed and improved the manuscript.

7. PLOS authors have the option to publish the peer review history of their article (what does this mean? ). If published, this will include your full peer review and any attached files.

**Do you want your identity to be public for this peer review?** For information about this choice, including consent withdrawal, please see our Privacy Policy .

Reviewer #1: **Yes: ** Nasser M Alorfi

---

## [Editor Report · Acceptance letter]

PONE-D-25-01650R1

PLOS ONE

Dear Dr. Almoghairi,

I'm pleased to inform you that your manuscript has been deemed suitable for publication in PLOS ONE. Congratulations! Your manuscript is now being handed over to our production team.

Kind regards,

on behalf of

Dr. Shukri AlSaif

Academic Editor

PLOS ONE